# CONIC: Contour Optimized Non-Iterative Clustering Superpixel Segmentation



Cheng Li [1], Baolong Guo [1,*], Nannan Liao [1], Jianglei Gong [1,2], Xiaodong Han [2], Shuwei Hou [1,2], Zhijie Chen [1] and Wangpeng He [1]

1   Institute of Intelligent Control and Image Engineering, Xidian University, Xi'an 710071, China; licheng812@stu.xidian.edu.cn (C.L.); nnliao@stu.xidian.edu.cn (N.L.); gongjianglei@stu.xidian.edu.cn (J.G.); swhou521@stu.xidian.edu.cn (S.H.); chenzhijie@stu.xidian.edu.cn (Z.C.); hewp@xidian.edu.cn (W.H.)
2   China Academy of Space Technology, Beijing 100094, China; willingdong@163.com
*   Correspondence: blguo@xidian.edu.cn; Tel.: +86-180-9280-1271

**Abstract:** Superpixels group perceptually similar pixels into homogeneous sub-regions that act as meaningful features for advanced tasks. However, there is still a contradiction between color homogeneity and shape regularity in existing algorithms, which hinders their performance in further processing. In this work, a novel Contour Optimized Non-Iterative Clustering (CONIC) method is presented. It incorporates contour prior into the non-iterative clustering framework, aiming to provide a balanced trade-off between segmentation accuracy and visual uniformity. After the conventional grid sampling initialization, a regional inter-seed correlation is first established by the joint color-spatial-contour distance. It then guides a global redistribution of all seeds to modify the number and positions iteratively. This is done to avoid clustering falling into the local optimum and achieve the exact number of user-expectation. During the clustering process, an improved feature distance is elaborated to measure the color similarity that considers contour constraint and prevents the boundary pixels from being wrongly assigned. Consequently, superpixels acquire better visual quality and their boundaries are more consistent with the object contours. Experimental results show that CONIC performs as well as or even better than the state-of-the-art superpixel segmentation algorithms, in terms of both efficiency and segmentation effects.

**Keywords:** superpixel segmentation; image contour; non-iterative clustering; seeds redistribution; feature distance measurement

## 1. Introduction

Superpixel segmentation is an important branch of image segmentation and multi-scale representation. It is essentially a process that over-segments an image into numerous spatially connected regions of similar size. Within it, a superpixel is a homogeneous description of texture, color or other features in accordance with visual perception, which substitutes for pixel-level features in object modeling. Since being introduced by Ren et al. [1] in 2003, research on superpixel segmentation gradually has become fundamental and popular in image processing and pattern recognition domains. Many advanced computer vision applications are developed on several outstanding superpixel generation algorithms to achieve more desirable results. Among them are target detection [2], object classification [3], image decomposition [4] and hierarchical representation [5]. In those works, superpixel segmentation performs as an efficient pre-processing step to compute image features and significantly reduces the number of entities in the following steps.

The pursuit of prominent superpixels is still a hotspot in this field from its birth to the present [6]. In recent years, a growing number of superpixel algorithms have been proposed to improve the segmentation performance [7]. Generally, the segmentation quality of superpixels is evaluated by accuracy, uniformity, compactness and time efficiency. Many state-of-the-art methods compute an eligibly balanced trade-off between these properties

via various techniques [8–13]. Typically, Simple Linear Iterative Clustering (SLIC) [9] acts as an enlightening pioneer to generate desirable superpixels, which has been extended in several latest works [14–21].

### 1.1. Drawbacks on Clustering Framework

In the practical applications, however, some drawbacks are exposed in SLIC due to the structure defect by its simple framework [18]. The following deficiencies and shortcomings are considered to be bottlenecks of performance:

- It utilizes a kind of unstable local K-means in clustering, which is susceptible to cluster initialization [22].
- It merely relies on local color features and makes a fixed trade-off with spatial distances to enforce the shape regularity [19].
- Redundant eigenvalue computations in overlapping local regions are repeated in several iterations [14].
- A split-and-merge heuristic is necessary for region connectivity, which is usually implemented by a connected components algorithm [17].

To further address the abovementioned limitations within SLIC-like methods, Achanta et al. [18] proposed the Simple Non-Iterative Clustering (SNIC) algorithm in their follow-up works. As the name asserts, SNIC works in a non-iterative manner and removes the limitations of SLIC that iterative label updating processes result in redundant creations of all elements in each restricted region. In SNIC, each pixel is inspected four times at most to calculate the feature distance. It adopts a priority queue to sort the inspected neighboring elements of all seeds, which substantially performs a label expansion process. Once a new label is assigned, the pixel will no longer be revisited. Consequently, much redundant computation for label updating can be prevented in overlapping local regions. Moreover, since all superpixels are generated by seeds expanding that absorb surrounding pixels, they still maintain the spatial connectivity of homogeneous pixels with the same labels. Therefore, the split-and-merge post-processing is omitted in SNIC.

Compared with SLIC, another important modification of SNIC is to generate superpixels whose clustering barycenters are evolved using online averaging that thoroughly avoids the iteration. Nevertheless, SNIC adopts a rigid region growing method to generate superpixels, in which a SLIC-like color-spatial feature distance is calculated. Thus, it may suffer from the shape compactness that goes against the local homogeneity. As a result, like other SLIC-like algorithms, it sometimes fails to adhere to image contours accurately, especially in complicated and textured regions.

### 1.2. Feasible Optimizations for Superpixels

As mentioned above, there is a large amount of literature on improving the performance of superpixels using various strategies. More subtle distance measurements and cost functions [23], elaborate feature spaces [17], as well as valuable prior information [24] are utilized in many state-of-the-art superpixel generation methods. Among them is gradient or contour prior information, which significantly avoids the crossing of image boundaries when associating a pixel to a superpixel. Giraud et al. [19] put forward a novel framework that provides a desirably balanced trade-off among segmentation quality and other characteristics of superpixels. The proposed Superpixels with Contour Adherence using Linear Path (SCALP) takes both regional color feature and contour intensity of all pixels on a linear path from a pixel to a cluster barycenter. Therefore, a joint color-spatial-contour homogeneous measurement has emerged. The resultant superpixels not only show regularity in sizes and shapes but follow regional color homogeneity.

It is also worth noting that, for seed-demand algorithms, a good initialization of seeds plays an important part in generating desirable superpixels [17]. Specifically, it is beneficial in the following three aspects:

- The clusters converge rapidly since the adjacent pixels are quite similar to the seed in each restricted region [18];

- More homogenous superpixels can be generated because the cluster barycenters avoid falling into local optimum [25];
- The resultant superpixels are more sensitive to image content, owing to the information-aware distribution of seeds [26].

Nevertheless, most approaches produce the initial seeds by regularly sampling on the image grid, thereby creating an even distribution of local information for all rectangular cells. As a bottom-up implementation based on data-driven, initialization in this way does not concern any varying content of the real-world images. Therefore, it results in inferior segmentation performance from the start, even though the positions of the seeds can be updated by the subsequent regional iterations. Moreover, since superpixels are often generated to speed up the subsequent visual analysis, the algorithm should perform efficiently in various practical tasks. Whereas in practice, the additional feature calculations on pixels along the linear path dramatically increase the computational cost, which severely limits its application.

Consequently, in the literature, each superpixel algorithm has its own bright spots and shortcomings. It is still challenging to look for the potential optimization that provides the best-balanced trade-off for particular applications. Theoretically, an outstanding superpixel segmentation algorithm could run sufficiently fast and provide the perfect segments. For example, well adherence to object boundaries or contours, as well as color homogeneity for complicated texture and small size regions. Moreover, for better visual quality, superpixels are expected to be compact, placed regularly and exhibit smooth boundaries [7].

### 1.3. Contributions of the Proposed Work

To achieve the aforementioned properties together, this paper proposes a new superpixel segmentation method, referred to as Contour Optimized Non-Iterative Clustering (CONIC). Enlightened by the structural properties of SNIC and SCALP, CONIC incorporates contour constraint into the non-iterative clustering (NIC) framework. In the initialization step, a redistribution strategy is performed that iteratively relocates and produces/eliminates the incipient seeds by color and contour similarity after uniformly grid sampling. This is done to achieve a better initial distribution for all seeds that start from homogeneous areas and avoid being located on the object edges or noisy points. During the joint online assignment and updating step, a subtle distance measurement is introduced to depict the similarity of a pixel with a cluster. Unlike the empirically fixed factor normalizing color and spatial proximity in many SLIC-like approaches, the difference of contour intensity works synthetically with color variation. The proposed measurement globally magnifies the feature distance so as to classify pixels in a more accurate manner, with emphasis on the weak boundary and textured regions. It also preserves a moderate spatial constraint that facilitates the visual perception of shape uniformity.

To sum up, the main contributions of this paper are:

- A new seed initialization strategy is introduced for the NIC framework. It efficiently overcomes the limitations of grid sampling via a global redistribution based on contour prior. As a result, it avoids clustering falling into local optimum, thus further generating more desirable superpixels with exactly the same number required by the user.
- A novel distance measurement is proposed to depict the similarity of a pixel with a cluster more accurately. It takes color information, contour prior and spatial constraint into consideration in a subtle way. Accordingly, the homogeneity and shape regularity of superpixels can be enhanced effectively, without deteriorating other characteristics.
- The proposed CONIC inherits both the efficiency of the NIC framework and the accuracy of contour-based distance measurement. Compared with SNIC, SCALP and other six state-of-the-art methods on a quantitative benchmark, it can generate comparable superpixels with respect to segmentation accuracy and visual effects in a limited computational time.

This paper is organized as follows. The background and related works are reviewed in the next section. In Section 3, the proposed CONIC method is presented in detail. Qualitative and quantitative results and discussion are explicated in Section 4. Section 5 displays the potential applications. Finally, Section 6 makes a brief conclusion and prospect.

## 2. Backgrounds

In this section, a rough categorization is adopted to introduce some representative state-of-the-art works. It classifies superpixel algorithms into seed-demand methods and graph-based approaches by the way of superpixel generation [27]. Moreover, the methodology of the NIC framework is presented at the end of this section.

### 2.1. Seed-Demand Superpixel Segmentation

The seed-demand methods, which are data-driven, tend to utilize a number of preset seeds to expand superpixels on the image plane with or without prior knowledge. Clustering-based, watershed-based and morphology-based approaches are mainly included in this category.

**Clustering-based**. Clustering-based methods usually generate superpixels by K-means that generate superpixels from the initialized seeds. Color and spatial information are generally utilized as the regional features. The abovementioned SLIC [9] is considered to be instructive for the following two aspects. First, the joint color-spatial space distance measurement could both control the size and compactness of superpixels. In addition, it localizes the search region for K-means clustering that globally avoids performing redundant distance calculations. Those two properties make SLIC appropriate for deployment and expansion in the follow-up works.

Linear Spectral Clustering (LSC) [16] and Intrinsic Manifold SLIC (IMSLIC) [17] extend SLIC by introducing k-means clustering into elaborately designed distance measurements and feature spaces. LSC applies a weighted k-means method in feature space with higher dimensions by kernel function, which reduces the time complexity of normalized cut (NC) [10] to linear and preserves the global image properties. By computing the geodesic centroidal Voronoi tessellation (GCVT) on a 2-dimensional manifold feature space, IMSLIC makes superpixels sensitive to image content without post-processing to enforce connectivity.

Preemptive SLIC (preSLIC) [14] and Fast Linear Iterative Clustering (FLIC) [15] are two representative accelerated optimizations of SLIC. In preSLIC, the deviation of cluster barycenters during each iteration is adopted as a homogeneity criterion to guide the convergence of candidate regions. Compared with conventional SLIC, only some deviation-variable superpixels are checked by the local K-means method in the updating step. Therefore, much redundant revisiting computation can be prevented. A major insight of FLIC can be generalized as a trade-off between shape uniformity and time efficiency. It assumes that neighboring pixels have natural continuity that tends to be assigned the same label. Based on the active search strategy by relationships between neighboring pixels, FLIC takes place in fixed search ranges in SLIC and achieves rapid convergence of linear clustering.

**Watershed-based**. Superpixels are essentially a special case of an image over-segmentation [1]. Therefore, some marker-controlled watershed transformations can also be classified into superpixel generation methods. Among them are Compact Watershed (CW) [14], Watershed Superpixels (WS) [28] and Waterpixels [29], which select several markers similar to sampling seeds in SLIC. Then, the watershed transformation is performed based on the makers and the image gradient, wherein a spatial constraint is introduced providing controllability over the compactness.

**Morphology-based**. Morphology-based algorithms generate superpixels as evolving outlines starting from initial seeds. In TurboPixels [8], a level-set-based geometric flow algorithm is adopted to dilate all seeds. It combines a curve evolution model for dilation with a skeletonization process for spatial constraint, thus generating highly regular superpixels. Topology Preserved Superpixel (TPS) [24] partitions an image into superpixels by connecting seeds through several optimal paths vertically and horizontally. A contour

prior is necessary for this method since the seed needs to be relocated to the pixel with locally maximal edge magnitudes. On this basis, Dijkstra's algorithm is used to generate the optimal path. Intuitively, TPS can generate topology preserved regular superpixels which are topology preserved.

### 2.2. Graph-Based Superpixel Segmentation

A graph-based superpixel algorithm generally produces superpixels via a graph model to depict the relationships between adjacent pixels in an image. In this category, graph cut, boundary evolution and energy optimization are three mainstreams [30].

**Graph cut**. Normalized Cuts (NC) [10] is a pioneering algorithm used in [1] to partition an image into regular and compact regions. In this method, each pixel is regarded as a node, and then the superpixel generation task is converted into recursively cutting the pixel graph thus minimizing a cost function based on contour and texture information. Entropy Rate Superpixel (ERS) [11] proposes an objective function based on the entropy rate of a random walk on the graph topology. It uses a priority queue to obtain edges to a new graph and calculates the entropy rate from the cut costs on the graph until the connected area reaches the expected number.

**Boundary evolution**. Superpixel Lattices (SL) [31] generates superpixels by constructing the vertical and horizontal superpixel boundaries. The optimal paths in both vertical and horizontal are searched within the predefined strips, which are then utilized to split an image and yield superpixels. Superpixels Extracted via Energy-Driven Sampling (SEEDS) [32] generate superpixels by iteratively evolving each initial rectangular region using coarse-to-fine pixel exchanges with neighboring superpixels. It adopts a hill-climbing algorithm to optimize an energy function formed by the histogram features of superpixels.

**Energy optimization**. Compact Superpixels (CS) and Constant-Intensity Superpixels (CIS) [33] are two approaches that formulate the superpixel segmentation problem in energy optimization. CS assumes that the input image is intensively covered by half-overlapping square patches of the same size, thus it shows uniform compactness with regular shape and size. CIS assigns each patch the color of the pixel at the center where every single pixel belongs to one of the overlapping patches. By adding a constraint to the energy function, the resultant superpixels have constant intensity. Lazy Random Walk (LRW) [34] converts segmentation into graph partition. The vertex of the graph is the image pixel and the edge is defined on the Gaussian weighting function. It iteratively optimizes superpixels by an energy optimization function based on texture information and object boundaries.

### 2.3. Preliminaries on Non-Iterative Clustering

Superpixel generation in a one-pass fashion has gained more attention in recent years [35]. Compared with the iteration-needed methods mentioned above, it fundamentally reduces the computational cost owing to the non-iterative structure. Since the proposed work attempts to combine contour prior with the NIC as a new framework to generate superpixels, the conventional principle, as well as some key notations in SNIC, is reviewed as follows:

- **Step 1**. In an image plane $I = \{I_i\}_{i=1}^N$, several pixels are sampled as the incipient seeds $\{s_k\}_{k=1}^K$ as well as the cluster barycenters $\{b_k\}_{k=1}^K$ with a unique label $L(s_k) = L(b_k) = k$. A small-root priority queue $Q$ is initialized that always returns the minimum key value while it is not empty.

- **Step 2**. For each seed $s_k$, the 4-neighboring unlabeled elements $\{I_j\}_{j=1}^4$ are inspected clockwise. The distance $D(I_j, b_k)$ is individually computed as the key value for each element $I_j$ before it is pushed on $Q$.

$$D(I_j, b_k) = \|C(I_j) - C(b_k)\|_2^2 + \|P(I_j) - P(b_k)\|_2^2 \cdot \left(\frac{N_{color}}{N_{spatial}}\right)^2, \tag{1}$$

where $C()$ is the color feature in 3-channel CIELAB space, $P()$ is the coordinate in 2-dimensional Euclidean space. $N_{color}$ and $N_{spatial}$ are two constants that represent the maximum color and spatial difference within the cluster $\Omega_k$. $\| \|_2$ represents the Euclidean distance. In addition, a joint color-spatial 5-dimensional feature $F()$ that describes $\Omega_k$ specifically

$$F(\Omega_k) = F(b_k) = \sum_{i \in \Omega_k} [C(I_i), P(I_i)] / |\Omega_k|, \tag{2}$$

where $| |$ means the number of pixels in a cluster region.

- **Step 3**. In the priority queue $Q$, the top-most element $I_q$ is popped and assigned a label $k$ identical to its seed $I_p$ that previously inspects $I_q$, i.e., $L(I_q) = L(I_p) = k$

$$I_q = \arg\min D(I_i, b_k), \ I_i \in Q, k \in [1, K]. \tag{3}$$

Then it updates the cluster $\Omega_k$ by

$$F(\Omega'_k) = F(b'_k) = \frac{\sum_{i \in \Omega_k} [C(I_i), P(I_i)] + [C(I_q), P(I_q)]}{|\Omega_k| + 1}, \tag{4}$$

where $\Omega'_k$ centered at $P(b'_k)$ is the updated cluster $\Omega_k$ that absorbs $I_q$ as a new element.

- **Step 4**. In the next loop, $I_q$ becomes the new seed of cluster $\Omega'_k$, and the 4-neighboring unlabeled elements of $I_q$ are processed similar to in Step 2.
- **Step 5.** Repeat Step 2 to 4 until $Q$ is empty.

Note that an inspected element may not be unique in $Q$. It would be revisited by other seed pixels with different labels, and calculated for more than one distance value. In this case, its final label is decided by the cluster with the minimum distance. A more illustrated procedure can be found in the literature [18,36].

## 3. Methods

As shown in Figure 1, the primary insight of the proposed CONIC can be generalized into three aspects. Firstly, it introduces contour prior to redistribute the grid sampled seeds, enabling them more adaptive to the image content (Figure 1c). Meanwhile, an improved feature distance is elaborated to measure the color similarity, which adopts contour constraint to multiply the feature distance of boundary pixels (Figure 1d). During the NIC, the joint feature distance could measure the relationship of a cluster barycenter and unlabeled elements more stably. As a result, CONIC superpixels could acquire better boundary adherence and visual quality (Figure 1e).

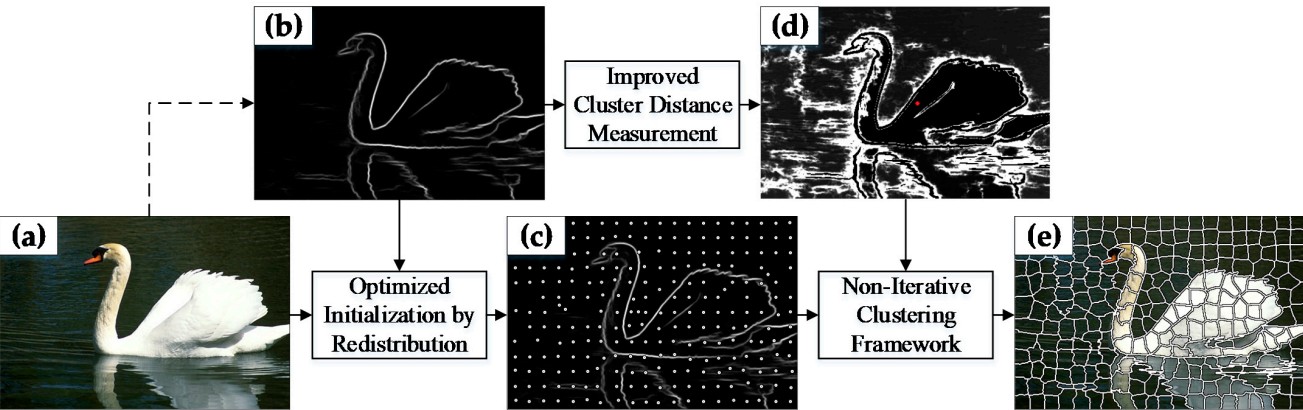

**Figure 1.** Scheme of the proposed Contour Optimized Non-Iterative Clustering (CONIC) superpixel segmentation method. (**a**) Input image; (**b**) Contour prior of (**a**); (**c**) Redistribution result of grid sampling; (**d**) Joint feature distance of (**a**); (**e**) Segmentation result (the boundaries of disjoint superpixels are graphically drawn by white curves on (**a**)). For visualization, the values are normalized in the range of 0 to 255 in (**d**), where whiter pixels indicate greater distances from a specific seed (red dot).

Moreover, as a plug-and-play role, any efficient contour detection method can be directly considered in Figure 1b (See [37] and references therein for more information, and this paper will not expatiate it). The detailed processes are discussed in the following subsections.

### 3.1. Optimized Initialization by Redistribution

As previously stated, SNIC is still not perfect, while it structurally overcomes some drawbacks in conventional SLIC. The most immediate one is the primitive grid sampling in seed initialization. The primary goal is to produce approximately the same size superpixels as the expected number. Nevertheless, it fails to concern varying content in the input image. For example, in some textured regions, it is difficult for superpixels to catch the boundaries accurately if the compactness constraint is too strong. Moreover, a detailed region probably needs densely distributed seeds so that the content can be partitioned homogeneously. Worse still, the rigid square grid cell constraint usually prevents actual generated superpixels from achieving the exact number of user-expectation [26].

To quickly generate a good seed initialization rather than straight evenly distribution, an efficient redistribution strategy is supplemented in this process. As shown in Figure 2, the core is to locally relocate all seeds followed by repeatedly modifying the total number based on the local color and contour similarity. The first step aims to create a rough distribution of seeds that avoid being located on the object edges or noisy points. Based on the inter-seed correlations, in the follow-up steps, a closely adjacent seeds pair would be merged into one. In another case, a new seed would emerge between a distant pair. After several instances of modification, the final distribution could work both on content-aware distribution and user-specified amount.

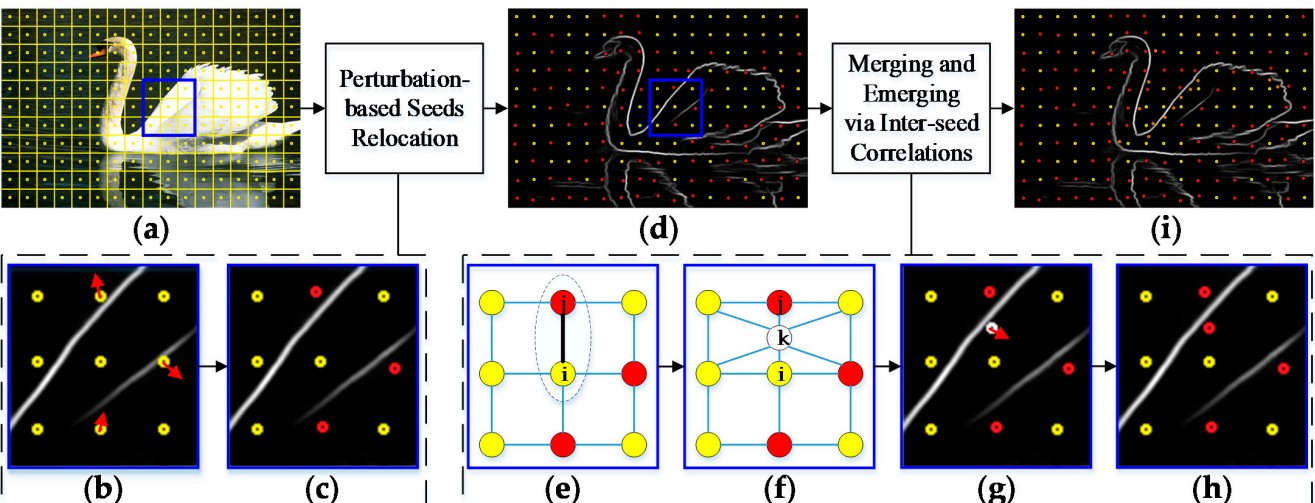

**Figure 2.** Dynamic redistribution procedure. (**a**) Grid sampled seeds; (**b**,**c**) Zoom-in perturbation-based seeds relocation in the blue rectangle of (**a**) (red arrows denote the motion of some seeds, and red seeds denote the relocated seeds); (**d**) Global perturbation results of all seeds. (**e**–**h**) Zoom-in performance of seeds redistribution, wherein a white-marked point denotes a newly merged/emerged seed in (**g**). (**i**) Result of optimized initialization. Another case that the actual number exceeds the expectation is shown in Figure 1c, which exhibits a consequence of seeds merging.

To begin with, in Figure 2a, the conventional grid sampling initialization is performed on the image plane with the expected superpixel number $K$. Theoretically, the initial seeds are evenly sampled, dividing the image into several square cells with the step of $S = \sqrt{N/K}$. Nevertheless, the actual side length of each cell $\lfloor S \rfloor$ (floor operation) is not always equal to $S$. It not only results in non-square cells in the image border but alters the resultant number of sampled seeds into $K'$.

Aiming at the incipient seeds $\{s_k\}_{k=1}^{K'}$ by grid sampling, firstly, a local relocation is performed on each seed $s_k$ within its corresponding cell $\square_k$. In Figure 2b,c, it is essentially a

perturbation process for all seeds that dynamically adjust themselves to the lowest positions of contour intensity within a range of $\lfloor S \rfloor \times \lfloor S \rfloor$. Accordingly, the seeds are relocated to $\{s'_k\}_{k=1}^{K'}$.

To make full use of the spatial relationship between the neighboring relocated seeds, as well as the intrinsic image features, inter-seed correlations are introduced to modify the number of seeds via several merging or emerging operations. Specifically, a region-based undirected graph $G = (V, E)$ is established to depict the correlations. Unlike many graph-based algorithms modeling all pixels, a node vertex $v_k \in V$ merely represents a perturbed seed $s'_k$ in this strategy. Meanwhile, edges $E \subseteq V \times V$ depict the correlation between two nodes $v_i$ and $v_j$ by

$$\omega_{ij} = \begin{cases} D_P(v_i, v_j) & \text{if } \square_i | \square_j \\ 0 & \text{otherwise} \end{cases}, \tag{5}$$

where $\square_i | \square_j$ means that the corresponding cells of $s'_i$ and $s'_j$ are spatially adjacent. In that case, the value of $\omega_{ij}$ equals a joint color-spatial-contour homogeneous measurement that firstly introduced in [19]

$$D_P(v_i, v_j) = d_{color}\left(s'_i, s'_j, \mathcal{P}_{ij}\right) \cdot d_{contour}\left(s'_i, s'_j, \mathcal{P}_{ij}\right) + d_{spatial}\left(s'_i, s'_j\right) \cdot \frac{m^2}{S^2}, \tag{6}$$

where $m$ is a user-specified compactness parameter. $d_{spatial}\left(s'_i, s'_j\right)$ is the spatial distance from $s'_i$ to $s'_j$ in the image plane

$$d_{spatial}\left(s'_i, s'_j\right) = \|P(I_j) - P(s_k)\|_2^2. \tag{7}$$

In the first term, $d_{color}\left(s'_i, s'_j, \mathcal{P}_{ij}\right)$ is an improved color distance that takes all pixels along the linear path $\mathcal{P}_{ij}$ from $s'_i$ to $s'_j$ into consideration

$$d_{color}\left(s'_i, s'_j, \mathcal{P}_{ij}\right) = \lambda \| C(s'_i) - C\left(s'_j\right) \|_2^2 + (1 - \lambda)\frac{1}{|\mathcal{P}_{ij}|} \sum_{k \in \mathcal{P}_{ij}} \| C(I_k) - C\left(s'_j\right) \|_2^2. \tag{8}$$

Similarly, $d_{contour}\left(s'_i, s'_j, \mathcal{P}_{ij}\right)$ accumulates the contour intensity on $\mathcal{P}_{ij}$ as

$$d_{contour}\left(s'_i, s'_j, \mathcal{P}_{ij}\right) = 1 + \frac{\gamma}{|\mathcal{P}_{ij}|} \sum_{k \in \mathcal{P}_{ij}} \left(1 - \exp\left(-c(I_k)^2/\sigma^2\right)\right), \tag{9}$$

where $c(I_k) \in [0, 1]$ is the normalized intensity of $I_k$ in the contour map. $\gamma$ and $\sigma$ are two parameters that weight the influence of the contour prior along $\mathcal{P}_{ij}$.

By calculating the edge weight $\omega_{ij}$, local information between adjacent seeds $s'_i$ and $s'_j$ can be roughly described. That is, if $\omega_{ij}$ is small enough, it indicates that the spatial distance is close and contour variation is not obvious. On the contrary, it would be relatively large if there exist contours between $s'_i$ and $s'_j$. Therefore, the homogeneity of $\square_i$ and $\square_j$ can be quantized.

Considering the above correlations, as well as the inequality of $K$ and $K'$, a priority queue $Q$ is adopted to iteratively modify the total number of seeds, which adopts the edge weights as the key value for sorting. As shown in Figure 2e, if the actual grid sampled seeds are insufficient compared with the expectation, the complementary seeds are resampled from the image plane. For a newly emerged seed $s_k$ in Figure 2f, it starts from the midpoint of an edge that holds the globally maximal weight. On the other hand, when the actual number exceeds the expectation, the seed pair with the globally minimal weight is merged to one that relocated in their spatial midpoint. For obtaining the corresponding extremum of $\omega_{ij}$, a big-root priority queue $Q_{\max}$ or a small-root one $Q_{\min}$ is adopted in the emerging

or merging operation, respectively. This process is iteratively performed until the actual number of seeds is adjusted to the user expectation.

Notably, any of the newly emerged or merged seeds from the above operations should move themselves to the lowest position of contour intensity (Figure 2g,h), which is similar to the incipient grid sampled seeds. In addition, since the seeds graph $G$ dynamically changes in the process, it should be updated after each operation, wherein the spatial relationship of the cell in which a new seed finally locates is utilized to modify the edge. Moreover, this procedure runs efficiently since there is a small difference between $K'$ and $K$ in general.

Eventually, an optimized distribution that is content-aware with the exact pre-set number of seeds is established. It is more sensitive to the complexity of regional information than conventional grid sampling, and also remains relatively far from the contour. This lays the foundation for efficient convergence without local optimum.

### 3.2. Improved Cluster Distance Measurement

In addition to the initialization, another bottleneck of the linear clustering framework is a trade-off between color consistency and spatial constraint. In other words, as for some SLIC-like superpixels, the shape regularity and size uniformity preserved by the second term in Equation (1) sometimes violates the color content in real-world images. As a result, some weak boundaries are difficult to adhere to, resulting in heterogeneous partitions.

Theoretically, the maximum spatial distance $N_{spatial}$ within a cluster is expected to be the sampling step $S = \sqrt{N/K}$. Nevertheless, the maximum color $N_{color}$ is widely different from cluster to cluster and image to image [9]. To handle this problem, all SLIC-like methods adopt an empirical constant $m$ as a substitute. In principle, the adherence of superpixel to image boundaries generally decreases with respect to a large $m$. On the contrary, a small $m$ would lead to irregular superpixels. In consequence, a reasonable value of $N_{color}$ for a balanced trade-off between segmentation accuracy and visual quality is hard to evaluate.

To further mitigate that contradiction in the NIC framework, an improved distance measurement is proposed to balance the abovementioned trade-off from a pixel $I_j$ to a cluster barycenter $b_k$, which is defined as

$$D'\left(I_j, b_k\right) = \left( \|C\left(I_j\right) - C\left(b_k\right)\|_2^2 + \|P\left(I_j\right) - P\left(b_k\right)\|_2^2 \cdot \left( \frac{N_{color}}{N_{spatial}} \right)^2 \right) \cdot N_{contour}, \quad (10)$$

$$N_{contour} = \exp\left(\varepsilon \cdot c\left(I_j\right)\right). \quad (11)$$

where $\varepsilon > 0$ balances the influence of the contour prior on the feature distance.

Enlightened by Equation (6) in SCALP, the local contour prior is recast into a variable coefficient in Equation (10), so that the final distance can be adjusted by a joint effort of color, space and contour. Compared with Equation (1), a new factor $N_{contour} \in [1, \exp(\varepsilon)]$ is introduced to adjust the feature distance from $I_j$ to $b_k$ along with $N_{color}$ and $N_{spatial}$. As shown in Figure 3c,d, if the color of an internal pixel is about the same as its neighboring elements with nearly zero contour intensity, the local region is supposed to be consistently smooth, wherein the improved feature distance from $I_j$ (blue) to $b_k$ (purple) maintains a low value in Figure 3e according to Equation (10). Since the color difference is negligible, the joint distance is mainly adjusted by the spatial term, which eliminates the over-smoothed effect and maintains a regular shape constraint. In addition, $N_{contour} \rightarrow 1$ keeps a stable multiplication of final $D'\left(I_j, b_k\right)$. On the other hand, if $I_j'$ (red) is a boundary pixel with a greater contour intensity, the corresponding $N_{contour}$ increases sharply, resulting in a local maximum value of $D'\left(I_j', b_k\right)$.

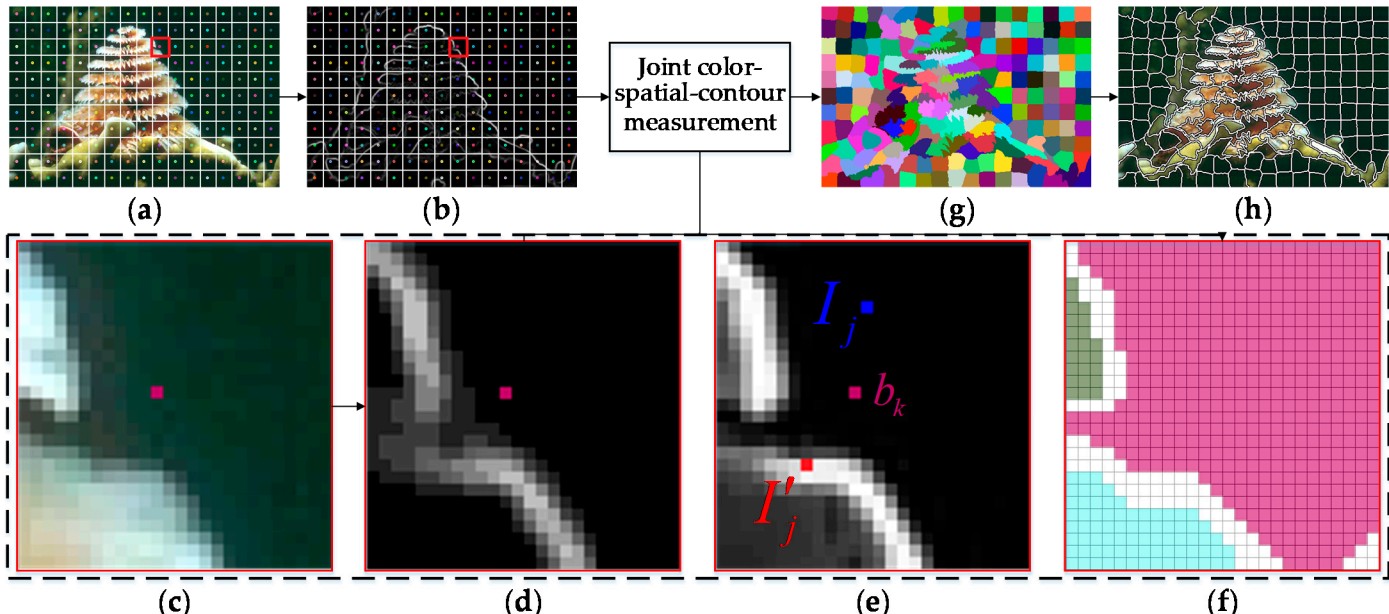

**Figure 3.** The principle of superpixel segmentation via the improved cluster distance measurement. (**a**,**b**) Grid sampled seeds distributed in the input image and its contour map (blacker pixels indicate smaller intensities), respectively; (**c**,**d**) Zoom-in performance of (**a**,**b**) in the red rectangle, respectively; (**e**) Local joint feature distance (whiter pixels indicate greater distances from $b_k$); (**f**) Label expansion in the NIC framework; (**g**) Result of label expansion; (**h**) Result of superpixel segmentation. The purple dot denotes a cluster barycenter $b_k$ in (**c–e**).

Considering that in the NIC framework, the distance measurement directly determines the feature similarity of a cluster and its neighboring pixels and the priority of label assignment. Therefore, in Figure 3f, contour pixels with greater distances tend to be assigned in the end, behind the other homogeneous elements. More generally, $N_{contour}$ increases monotonically with the contour intensity in Equation (11), thus prevents the boundary pixels from being assigned prematurely and guarantees accurate convergence of all clusters. As a result, the outlines of superpixels are more consistent with the object boundary. It is also worth noting that, as a label expansion process, the framework maintains spatial connectivity of the same labels [36]. It indicates that an unlabeled pixel would neither be inspected nor assigned by a cluster if there is a set of continuous contour pixels between them, which further avoids the updating bias of cluster barycenters and misclassification of more pixels.

Figure 4 illustrates a set of visual results that adopts a different kind of feature distances. Both SLIC and SCALP perform on linear iterative clustering framework, while SNIC and the proposed distance measurement work in a non-iterative manner. In Figure 4b, SCALP calculates the color and contour intensities of all pixels along the linear path. In Equation (6), the term $d_{contour}\left(b_k, I_j', \mathcal{P}_{kj}\right)$ could prevent a pixel $I_j'$ to be associated with the cluster centered at $b_k$ in the assignment step. Compared with SLIC based on Equation (1) in Figure 4a, the irregularity of the superpixel shape is ameliorated, along with better boundary adherence.

Based on conventional color-spatial distance, the proposed measurement achieves a more satisfactory performance. It becomes easier for clusters to classify homogeneous pixels accurately in the NIC framework (Figure 4d). Accordingly, superpixels in smooth regions maintain the compactness as the initial rectangle. Moreover, boundaries are more consistent with the object contours in detailed segments. Furthermore, unlike Equations (8) and (9) which take all pixels along the linear path into the calculation, the proposed measurement only concerns the feature of the current inspected pixel. In other words, it follows the measurement style of SNIC that inspects pixels individually, so that time efficiency can be guaranteed.

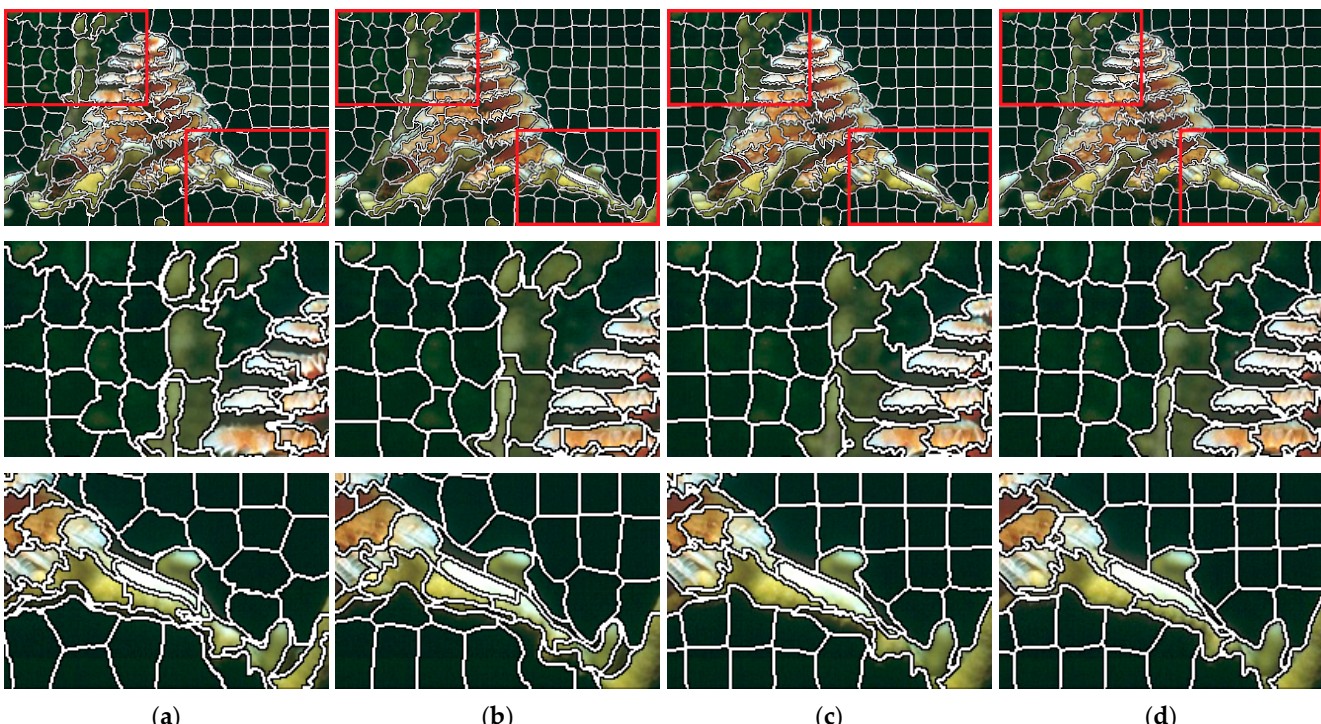

|  |  |  |  |
|:---:|:---:|:---:|:---:|
| (**a**) | (**b**) | (**c**) | (**d**) |

**Figure 4.** Visual results of superpixel segmentation. (**a**) Simple Linear Iterative Clustering (SLIC); (**b**) Superpixels with Contour Adherence using Linear Path (SCALP); (**c**) Simple Non-Iterative Clustering (SNIC); (**d**) NIC via the proposed distance measurement. Each segmented image is generated by nearly 200 superpixels, followed by two zoom-in results. See Section 4 for more comprehensive evaluations.

### 3.3. Synergetic CONIC Framework

Combining the seeds-optimized initialization and distance-improved clustering synergetically, there is a series of positive effects on the performance of the proposed CONIC superpixels:

- The optimized initialization strategy helps the improved distance measurement work better on the NIC, since all clusters should evolve from a flat location in the latter strategy. Otherwise, there would be some orphaned clusters with very small sizes.
- The optimized initialization strategy could reflect the distribution of image content, which facilitates the improved distance measurement dealing with some complex scenes, such as small objects and weak boundaries.
- The combination shows an immediate effect on improving the segmentation quality, and the parameters lead to very similar performance within a wide adjustment range. Therefore, it is easy to set a proper value for each parameter in these two strategies.
- The combination makes for a robust and rapid convergence of each cluster in the NIC, without concerning the boundary pixels. Potentially, these elements can be inspected only once which avoids calculating feature distance repeatedly.

Overall, CONIC inherits both the computation efficiency of the NIC and adherence to contour prior that produces desirable segmentation accuracy. Moreover, it shows a better visual quality that the shape and size of superpixels being consistent and regular. A pseudocode summary of the framework is presented in Algorithm 1.

---

**Algorithm 1:** CONIC superpixel segmentation framework

---

**Input:** the Lab image $I$, the expected superpixel number $K$, the normalized contour map $M$
**Output:** the label map $L$ of $I$
/* Initialization */
initialize cluster seeds $\{s_k\}_{k=1}^{K'}$ by grid sampling at a regular step $S$.
initialize a priority queue **Q** with a small root.
set label $L(I_i) = -1$ for $\{I_i\}_{i=1}^{N}$.
/* Seeds redistribution */
move $\{s_k\}_{k=1}^{K'}$ to the lowest positions of contour intensity within a range of $\lfloor S \rfloor \times \lfloor S \rfloor$.
create a region adjacency graph $G$ to depict the correlations of the perturbed seeds $\{s_k'\}_{k=1}^{K'}$.
**while** the number of nodes $N_g$ in $G$ is not equal to $K$ **do**
   **if** $N_g > K$ **then**
      calculate the midpoint $s_{mn}'$ of $s_m'$ and $s_n'$ corresponding to the minimum $\omega_{mn}$.
     move $s_{mn}'$ to the lowest positions.
      update $G$ that replace $s_m'$ and $s_n'$ with $s_{mn}'$.
   **else if** $N_g < K$ **then**
     calculate the midpoint $s_{ij}'$ of $s_i'$ and $s_j'$ corresponding to the maximum $\omega_{ij}$.
     move $s_{mn}'$ to the lowest positions.
     update $G$ that incorporate $s_{mn}'$.
   **end if**
  **end while**
/* Joint assignment and updating */
**for** each cluster barycenter $b_k$ in $\{b_k\}_{k=1}^{K}$ **do**
   create a vector node $[F(b_k), k, D'(s_k, b_k)]$.
   push the node on **Q** that adopts the distance $D'()$ as the key value for sorting.
**end for**
**while Q** is not empty **do**
   pop the top-most node $\left[F(I_q), k, D'(I_q, b_k)\right]$ corresponding to $I_q$ from **Q**.
   **if** $I_q$ is not labeled before **then**
     assign the label of $b_k$ to $I_q$.
     update the corresponding cluster by Equation (4).
     **for** each 4-neighboring element $I_p$ of $I_q$ **do**
       **if** $I_p$ is not labeled before **then**
         push the node $\left[F(I_p), k, D'(I_p, b_k')\right]$ on **Q**.
       **end if**
     **end for**
   **end if**
**end while**
return the label map $L$ of $I$

---

## 4. Results and Discussion

In this section, the proposed CONIC is evaluated to verify the superiority. The dataset and benchmark are introduced firstly, in which the qualitative and quantitative performance is tested and demonstrated from part to whole. After that, the computational efficiency is analyzed.

### 4.1. Experimental Setup

The experiments are performed on the Berkeley Segmentation Data Set 500 (BSDS500) [37], which contains 500 images with the size of $481 \times 321$ or $321 \times 481$. The proposed CONIC is compared with SNIC and SCALP to prove the superiorities, as well as other three seed-demand methods, namely TPS [24], SLIC [9] and WS [28]. In addition, three popular graph-based superpixel algorithms, SEEDS [32], ERS [11] and LRW [34] are also introduced as references. The other eight state-of-the-art methods are all based on available code with default parameters. CONIC is implemented by C/C++, wherein all parameters are set as in [38] except $\varepsilon$, which is set to 10 in this paper. Table 1 summarizes the properties of the algorithms in the experiments, including the category, controllability of superpixel number,

controllability of shape compactness, dependence of iteration, computational complexity and implementation. All methods are executed on an Intel Core i7 4.2 GHz with 16 GB RAM without any parallelization or GPU processing.

**Table 1.** Properties of the superpixel algorithms in the experiments. ●/○ means that the algorithm has/has not got the property.

| Methods | Seed-Demand | | | | | Graph-Based | | |
|---|---|---|---|---|---|---|---|---|
| | **SNIC** | **SCALP** | **TPS** | **SLIC** | **WS** | **SEEDS** | **ERS** | **LRW** |
| Number | ● | ● | ● | ● | ● | ● | ● | ● |
| Compactness | ● | ● | ○ | ● | ● | ○ | ○ | ● |
| Iteration | ○ | ● | ○ | ● | ○ | ● | ● | ● |
| Complexity | $O(N)$ | $O(N)$ | $O(N \log N)$ | $O(N)$ | $O(N)$ | $O(N)$ | $O(N \log N)$ | $O(N^2)$ |
| Code | C/C++ | C/Matlab | Matlab | C/C++ | C/Matlab | C/C++ | C/Matlab | C/Matlab |

In the following experiments, the performance of CONIC is evaluated quantitatively by the following four popular metrics in the field of superpixel segmentation [39]:

- Boundary Recall (BR). BR is the most commonly used metric to assess boundary adherence given ground truth. Mathematically, it is the ratio of ground truth boundaries covered by superpixel boundaries (higher is better).
- Under-segmentation Error (UE). UE utilizes segmentation regions instead of boundaries to penalize superpixels that overlap with multiple objects (lower is better).
- Achievable Segmentation Accuracy (ASA). ASA quantifies the accuracy achievable by following steps. A higher ASA value indicates the performance of superpixels in subsequent is unaffected (higher is better).
- Compactness (CO). CO refers to the area covered by individual superpixels that compares the area of each superpixel with the area of a circle (the most compact 2-dimensional shape) with the same perimeter (higher is better).

### 4.2. Algorithm Analysis

As mentioned above, the proposed CONIC consists of two main optimizations, which generate a synergetic effect on the clustering framework. In this subsection, the initialization-optimized SNIC (IO-SNIC) based on seed redistribution and the distance-optimized SNIC (DO-SNIC) based on an improved measurement that deploys one strategy on SNIC is implemented separately for comparison. In addition, the relations among SNIC, IO-SNIC, DO-SNIC and CONIC are pointed out.

#### 4.2.1. Visual Assessment

Figure 5 illustrates several subjective results for visual analysis of superpixels. Compared with conventional SNIC, IO-SNIC yields desirable performance in two aspects. First, it modifies some seeds from the initial evenly sampled position so that the superpixels can be avoided clustering from object contour or noisy points. Accordingly, superpixels are more aware of the image content that could promote boundary adherence, e.g., small objects with rich details. Moreover, the dynamic emerged or merged operations in IO-SNIC also render strict controllability of user-specified superpixel numbers. Compared to the first two methods, DO-SNIC exhibits strong shape consistency without apparent misclassification of homogeneous pixels. Moreover, it performs stably in both textured regions and smooth background that shows better visual effects.

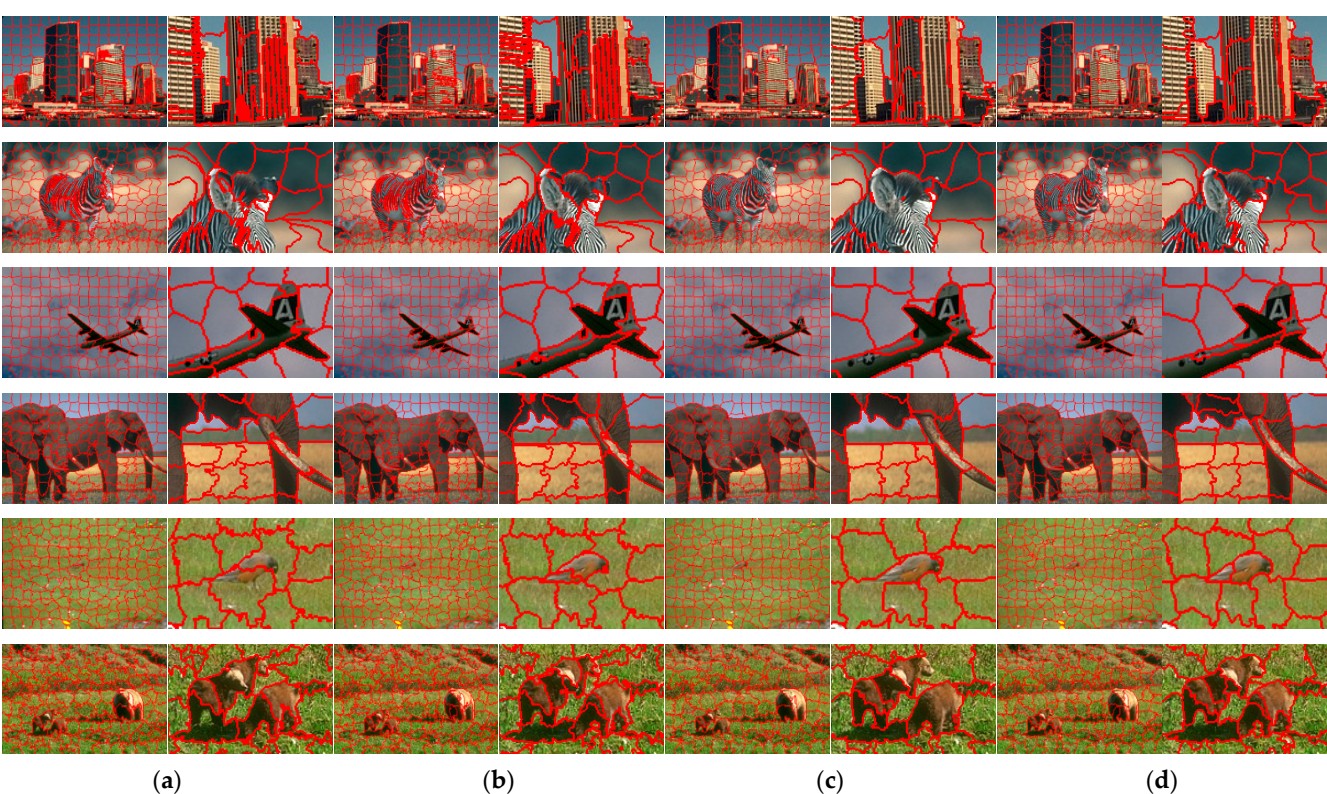

      (**a**)           (**b**)           (**c**)           (**d**)

**Figure 5.** Subjective results of superpixels on BSDS500. Each column represents superpixels generated by (**a**) SNIC; (**b**) initialization-optimized SNIC (IO-SNIC); (**c**) distance-optimized SNIC (DO-SNIC); (**d**) CONIC. The excepted number of superpixels is fixed to 200. Alternating columns show each segmented image followed by local details.

Owing to the intrinsic properties derived from IO-SNIC and DO-SNIC, the proposed CONIC could achieve more outstanding segmentation. Some low contrast fragments under-segmented by a single strategy are partitioned exactly, without large shape deformation. Moreover, superpixels become more simply connected, the evenly distributed seeds that might generate isolated clusters are redistributed in the very beginning. The joint color-spatial-contour homogeneous measurement provides both acceptable boundary adherence and shape stability. That is, CONIC acquires the best trade-off between segmentation accuracy and visual uniformity.

### 4.2.2. Metric Evaluation

Figure 6 quantitatively evaluates the influence of two strategies on the NIC framework via several metrics mentioned above. As to IO-SNIC, it is more remarkable to further promote the effectiveness of DO-SNIC rather than simply improve the performance of SNIC. Theoretically, DO-SNIC pursues strong edge support of superpixels while maintaining the shape regularity, thus achieving desirable results in terms of UE, ASA and CO. In general, a higher compactness constraint usually leads to lower boundary adherence in DO-SNIC. To ameliorate this problem, CONIC adopts the seeds redistribution strategy to generate superpixels from non-contour seed pixels via the optimized distance measurement. As shown in Figure 5c,d, the redistributed seeds that concern varying content could pay sufficient attention to local details. In addition, it also ensures a size uniformity of superpixel generation. Consequently, it is accepted that the proposed two strategies could generate a synergetic effect that optimizes the SNIC segmentation result desirably.

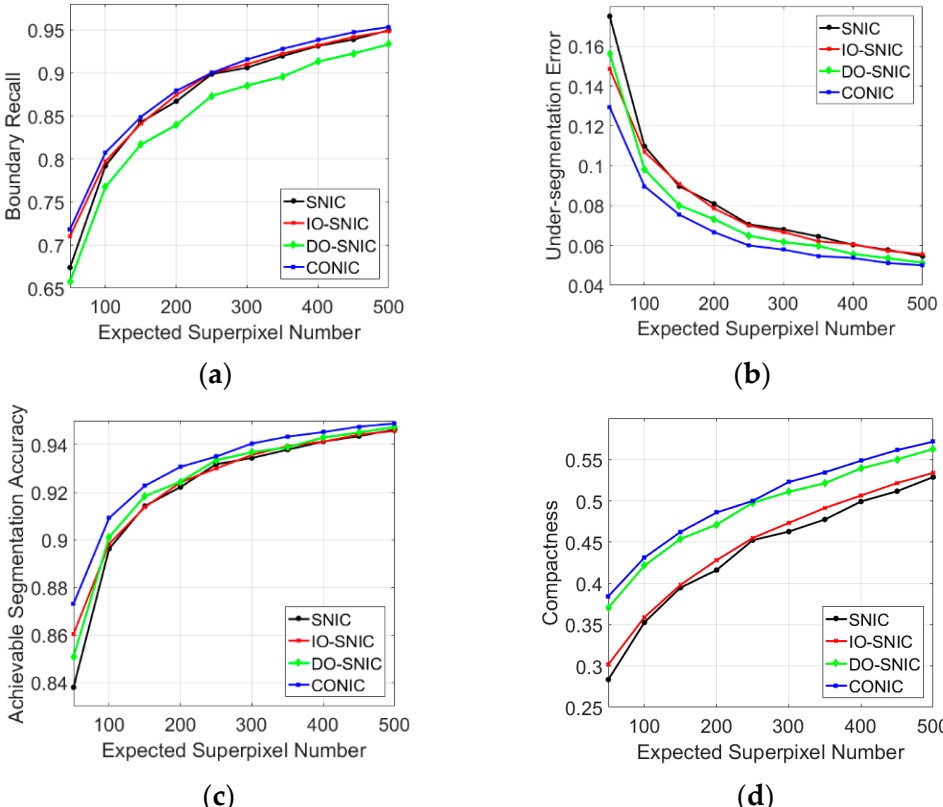

**Figure 6.** Quantitative evaluation of four algorithms on BSDS500. (**a**) Boundary recall; (**b**) Under-segmentation error; (**c**) Achievable segmentation accuracy; (**d**) Compactness. The expected number of superpixels ranges from 50 to 500.

### 4.3. Performance Comparison

As shown above, CONIC inherits the advantages of both IO-SNIC and DO-SNIC that could act synergistically in the NIC framework. To further verify its effectiveness, the proposed CONIC algorithm is roundly compared with methods in Table 1 (SNIC is excluded since it is individually analyzed with CONIC in the previous subsection).

#### 4.3.1. Comprehensive Evaluation

Figures 7 and 8 show the qualitative and quantitative performance of eight methods on BSDS500, respectively. It can be observed from Figure 7f,g that both SEEDS and ERS generate clutter and sinuous boundaries while the superpixels catch almost all details in each image, which makes them too chaotic to analyze. On the other hand, all seed-demand methods could generate regularly shaped superpixels including LRW, which also obtains superpixels by seed generation. Generally, these algorithms initialize a set of evenly distributed seeds on a regular grid, so that the produced superpixels tend to maintain approximately uniformity in shape and size. As an elaborately tailored framework for superpixel clustering, SLIC performs a balanced trade-off between shape regularity and boundary adherence. Nevertheless, there are still sinuous boundaries around some homogeneous superpixels. SCALP introduces a new distance measurement based on contour prior, whose resultant boundaries are smoother than SLIC to some extent. TPS also adopts contour information to generate superpixels. Other than linear iterative clustering of SLIC and SCALP, it aims to find an optimal path between each neighboring seed pairs that are relocated on the object boundaries. Owing to the shortest path vertically and horizontally, the TPS superpixels are topology preserved that seems very trim. However, it performs poorly in aligning to objects. WS introduces a SLIC-like spatial constrain in the watershed framework to perform over-segmentation. As a result, superpixels are uniform

in simple backgrounds and variable in objects. On the contrary, it still cannot adhere to the boundary very well since the shape constrain is too rigid.

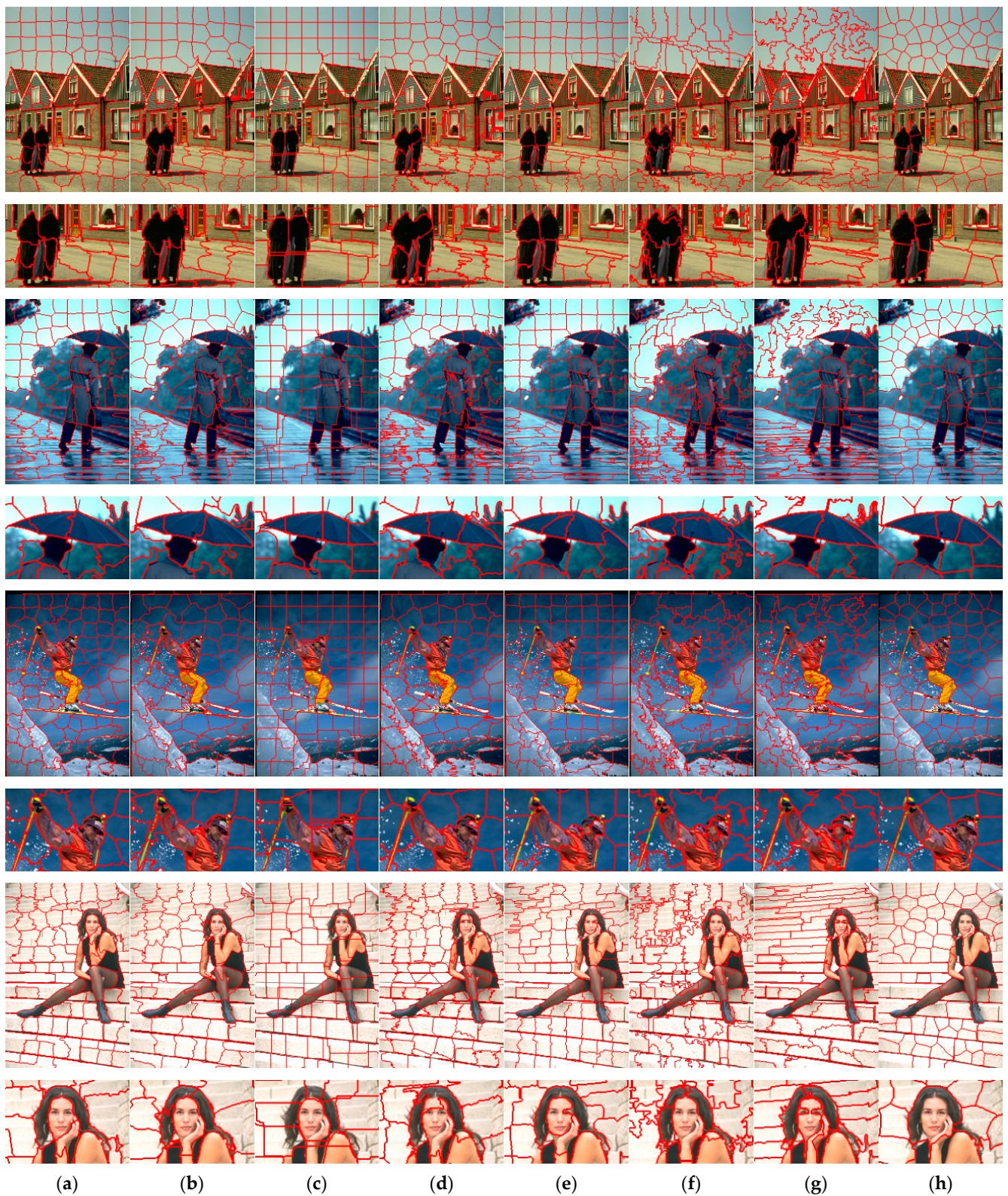

(**a**)          (**b**)          (**c**)          (**d**)          (**e**)          (**f**)          (**g**)          (**h**)

**Figure 7.** Visual comparison of segmentation results with 100 expected superpixels. (**a**) CONIC; (**b**) SCALP; (**c**) Topology Preserved Superpixel (TPS); (**d**) SLIC; (**e**) Watershed Superpixels (WS); (**f**) Superpixels Extracted via Energy-Driven Sampling (SEEDS); (**g**) Entropy Rate Superpixel (ERS); (**h**) Lazy Random Walk (LRW). Alternating rows show each segmented image followed by the zoom-in performance.

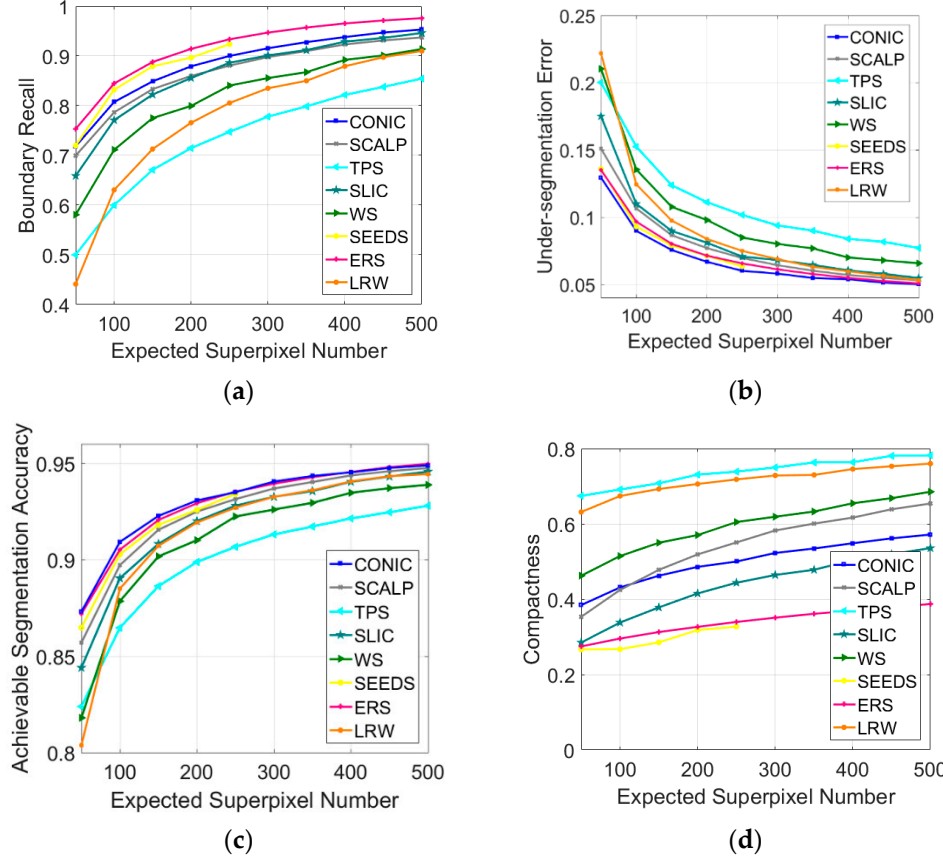

**Figure 8.** Quantitative evaluation of eight algorithms on BSDS500. (**a**) Boundary recall; (**b**) Under-segmentation error; (**c**) Achievable segmentation accuracy; (**d**) Compactness. The expected number of superpixels ranges from 50 to 500 (SEEDS is not plotted integrally due to its unsupported parameter).

Compared with those results, the proposed CONIC in Figure 7a shows better visual effects from whole to part. Firstly, it overcomes the instability in grid sampling initialization of SLIC and WS, in which the seeds are simply perturbed to the lowest gradient positions in its $3 \times 3$ neighborhood. The inter-seed correlations, established on contour prior, could both guide seeds to distribute in a content-aware manner and avoid them located on the image edge. It is also proved by LRW that the seed redistribution strategy could help the boundaries of final superpixels adhere to object boundaries more tightly. Furthermore, other than TPS or SCALP which merely adopts the contour prior in initialization or clustering, CONIC leverages as much of this information as possible. Besides the abovementioned seeds redistribution strategy, contour prior is also utilized in the distance measurement. It substitutes the rigid color-spatial by a joint color-spatial-contour distance in Equation (1) in SNIC. As a synergetic result, CONIC superpixels ensure boundary adherence and maintain a stably regular shape. Compared with SCALP that also optimizes the distance measure in a similar manner, the spatial constraint is more adaptive to image content and performs better in twig objects as well as textured areas.

Figure 8 plots the quantitative evaluation of all comparison methods. CONIC outperforms the others in terms of UE and ASA (Figure 8b,c). It indicates that the result superpixels are likely to overlap with only one ground truth object, which could perform better in other unsupervised image segmentation applications as well. As for BR that evaluates the degree of ground truth boundaries detected by the superpixels, it does not consider the false detection in theory. Thus CONIC lags behind ERS and SEEDS, since there are a large number of boundary pixels in the two kinds of extremely irregular superpixels. CO is another important indicator to assess the visual quality of superpixels. Although CONIC merely exceeds SLIC, ERS and SEEDS in Figure 8d, it still presents relatively clear and tidy segmentation results (Figure 8a). From another perspective, regularity and smoothness are

also crucial in visual assessment. For example, TPS exhibits high compactness but poor regularity, and LRW shows high compactness but inferior smoothness. In fact, these two superpixel algorithms are both inferior in segmentation performance.

In general, the performance of an algorithm improves when the superpixel density increases. Whereas several seed-demand methods need sufficient initial seeds to maintain eligible segmentation results. That is, a proper user-specified input parameter is important to the superpixel quality. Specifically, CONIC works stably through the entire range of user-expected superpixel numbers from 50 to 500 on BSDS500. It not only maintains superiority at lower densities but retards the slope of superpixel number for better results. Overall, it could become a desirable region-based feature extractor for advanced visual tasks without much concentration on setting the parameters.

It is also worth noting that, as the most related work to CONIC, SCALP makes a balanced trade-off among these metrics. Nevertheless, the split-and-merge post-processing in the SLIC framework restricts its performance, especially in twig segmentation. As mentioned in Figure 7, its distance measurement is not robust to texture, either. On the contrary, CONIC deploys the joint color-spatial-contour homogeneous measurement on the NIC more subtly, along with seed redistribution strategy. It overcomes several inherent shortcomings in SCALP and achieves higher BR, UE and ASA. Moreover, it can unite better regularity and smoothness together with compactness to promote the visual quality outstanding.

### 4.3.2. More Discussion

In addition to the four metrics listed above, there are another two aspects that should be taken into consideration. The input/output number of user-expected superpixels and the execution time (ET) usually affect the consequent applications.

Table 2 illustrates the numerical comparison of superpixels between user-expectation (input value) and actual generation (mean value). If they are equal, the corresponding method can be considered controlling superpixel numbers exactly. As can be observed, only CONIC and ERS could generate exactly the same number of superpixels required by the user. All other seed-demand algorithms merely adopt the grid sampling initialization in Figure 3a. As mentioned in Section 3.1, it usually adjusts the input number to fit the requirement of square cells. LRW places the initial seeds in a similar way, but the final amount after iterative refinement is out of control. Additionally in SLIC and SCALP, the split-and-merge post-processing step further impact the final results. SEEDS proposes a block of pixels at different sizes as the initial superpixels, which requires the input number to proportionate to image size. Whereas in practice, a wide range of expected input is not suit for SEEDS in BSDS500. ERS searches for a graph topology that has the specific number of connected subgraphs in a graph model so that it could partition an image into any number of superpixels. As for CONIC, it maintains the property of IO-SNIC that adjusts the number of seeds via several merging or emerging operations shown in Figure 2e–h.

**Table 2.** Comparison of superpixel number between user-expectation and actual generation on BSDS500. Merely CONIC and ERS (in blue) could generate exactly the same amount.

| Method | User-Expected Superpixel Number | | | | | | | | | |
|---|---|---|---|---|---|---|---|---|---|---|
| | **50** | **100** | **150** | **200** | **250** | **300** | **350** | **400** | **450** | **500** |
| **CONIC** | 50 | 100 | 150 | 200 | 250 | 300 | 350 | 400 | 450 | 500 |
| **SNIC** | 40 | 96 | 150 | 187 | 260 | 294 | 330 | 400 | 442 | 504 |
| **SCALP** | 45 | 88 | 136 | 181 | 226 | 275 | 317 | 367 | 415 | 454 |
| **TPS** | 54 | 96 | 150 | 204 | 247 | 294 | 345 | 384 | 442 | 486 |
| **SLIC** | 41 | 92 | 143 | 185 | 252 | 289 | 324 | 394 | 436 | 496 |
| **WS** | 40 | 96 | 150 | 187 | 260 | 294 | 330 | 400 | 442 | 500 |
| **SEEDS** | 50 | 100 | 150 | 200 | 266 | - | - | - | - | - |
| **ERS** | 50 | 100 | 150 | 200 | 250 | 300 | 350 | 400 | 450 | 500 |
| **LRW** | 40 | 99 | 153 | 204 | 258 | 311 | 361 | 417 | 470 | 513 |

Figure 9 shows the execution time of CONIC from whole to part. Figure 9a suggests that it obtains comparable running times to the state-of-the-art methods. It also shows that CONIC, SNIC, WS and SEEDS are the fastest, which run twice as fast as SLIC on average. On the other side, SCALP, TPS and ERS are an order of magnitude slower than the first echelon. Benefited from the NIC framework, CONIC could converge efficiently with $O(N)$ time complexity. Particularly, it avoids calculating the additional distance of pixels along the linear path in Equation (10) and therefore the computation cost is dramatically reduced. As a result, it runs over 15 times faster than SCALP with respect to a wide range of expected superpixel numbers. Specifically, in Figure 9b, the execution time of SNIC, IO-SNIC, DO-SNIC and CONIC are plotted, which explains the additional runtime spent on the NIC framework. As can be seen, compared with conventional SNIC, a little extra time is spent on deploying the proposed two strategies. Intrinsically, the additional time is almost spent on IO-SNIC that iteratively performs the seed redistribution. Even though, it still runs in near real-time and can be considered better than SNIC since it improves the segmentation quality significantly as shown in Figures 5 and 6.

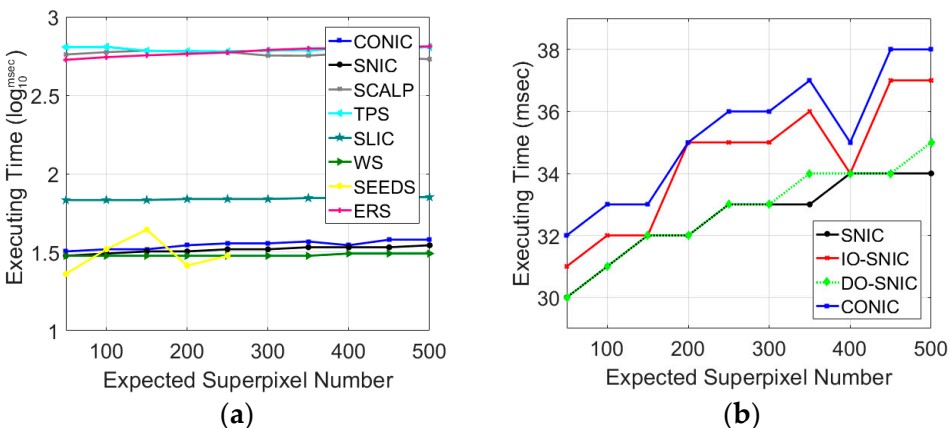

**Figure 9.** Comparison of execution time. (**a**) CONIC and other eight state-of-the-art methods (LRW is not plotted due to its relatively slow speed); (**b**) CONIC and one-strategy-implemented SNIC.

## 5. Applications

This section shows that superpixels generated by the proposed CONIC facilitate the application of remote sensing (RS) analysis and image segmentation. Since superpixel segmentation is generally adopted as a pre-processing step, the extracted superpixels are expected to improve the performance and efficiency of advanced tasks.

### 5.1. Multi-Resolution RS Imaging Segmentation

In order to demonstrate the availability in multi-resolution segmentation of RS images, CONIC superpixels are compared with the state-of-the-art algorithm [40] in eCognition v9.0, one of the most popular commercial RS software. In addition, this subsection follows the experimental style in [5] that mainly focuses on the comparison of time efficiency and visual quality.

As shown in Table 3, RS images from four different earth observation satellites are utilized with true color images in this experiment. The corresponding segmentation results are illustrated in Figure 10. By setting proper parameters, there is comparable performance between the two methods.

**Table 3.** Information and execution time of different segmented remote sensing (RS) images.

| Index | Image Information | | | | Execution Time (sec) | |
|---|---|---|---|---|---|---|
| | Satellite | Resolution | Image Size | Description | eCognition | CONIC |
| (a) | SPOT-5 | 2.5 m | $6000 \times 6000$ | Bandar-e Eman Khomeyn, Iran | 51.59 | 38.891 |
| (b) | Landsat-5 | 30 m | $5000 \times 4000$ | Blue algae eruption in Lake Erie, USA | 29.359 | 20.609 |
| (c) | TerraSAR-X | 1 m | $2880 \times 1440$ | Overlook of Harwell, Britain | 6.396 | 3.155 |
| (d) | WorldView-2 | 0.5 m | $1300 \times 1300$ | Varyag in the Yellow Sea, China | 5.913 | 1.3 |

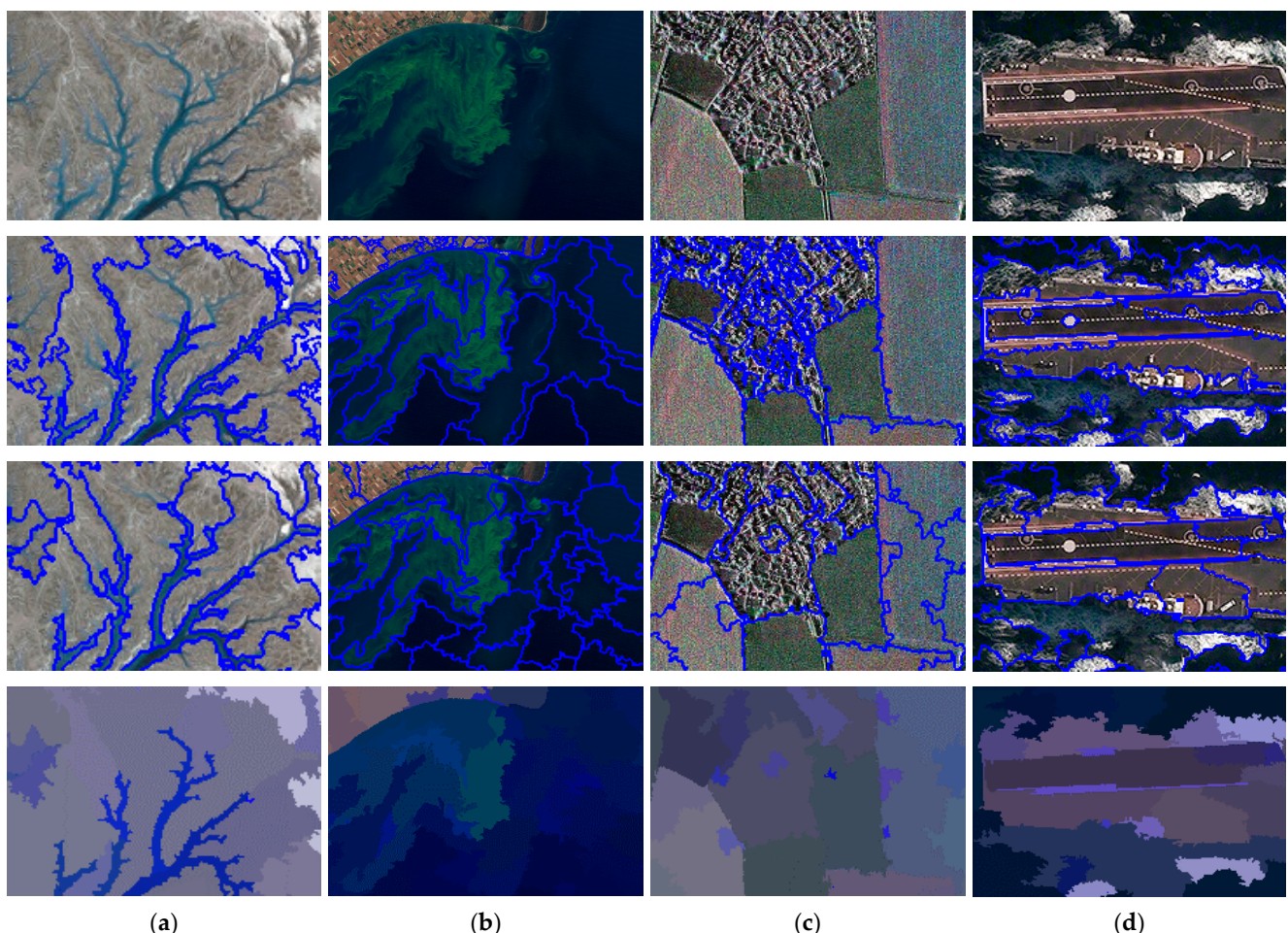

(**a**)　　　　(**b**)　　　　(**c**)　　　　(**d**)

**Figure 10.** Multi-resolution segmentation results of remote sensing images acquired by different satellites in Table 3. (**a**) SPOT-5; (**b**) Landsat-5; (**c**) TerraSAR-X; (**d**) WorldView-2. From top to bottom, input image, result of eCognition, result of CONIC, reconstruction from average colors on CONIC superpixels, respectively. Each image is cropped since the original size is too large to display the details. Note that the parameters in eCognition are fine-tuned to achieve approximate results with CONIC.

It is also worth noting that the segmentation results and execution time of eCognition are very sensitive to the segmentation settings. For example, there are three parameters in eCognition, scale, shape and compactness. A proper combination is widely different from image to image, resulting in unstable execution time. On the contrary, the proposed CONIC contains only one pre-set parameter, the expected superpixel number $K$ and runs with linear complexity in the number of image pixels $N$. Meanwhile, it could balance the visual perception that locally catches the details of land cover and globally maintains shape and size uniformity, with robustness in textured regions. As shown in the last row of Figure 10 (each image is computed by the average color on each superpixel), CONIC provides a much more visually satisfying reconstructed result and significantly reduces

the computational entities. Consequently, it would be an efficient tool for supervised land cover classification and object detection.

### 5.2. Pre-Process of Image Segmentation

Similar to the experiments demonstrated in [27], the achievable segmentation results are visualized in Figure 11, wherein the ground-truth labels are assigned to the superpixels whose elements mostly belong to a specific object class. Accordingly, the final label map can be regarded as the result of image segmentation based on an ideal classifier. Therefore, the original image with objects identified can be analyzed intuitively.

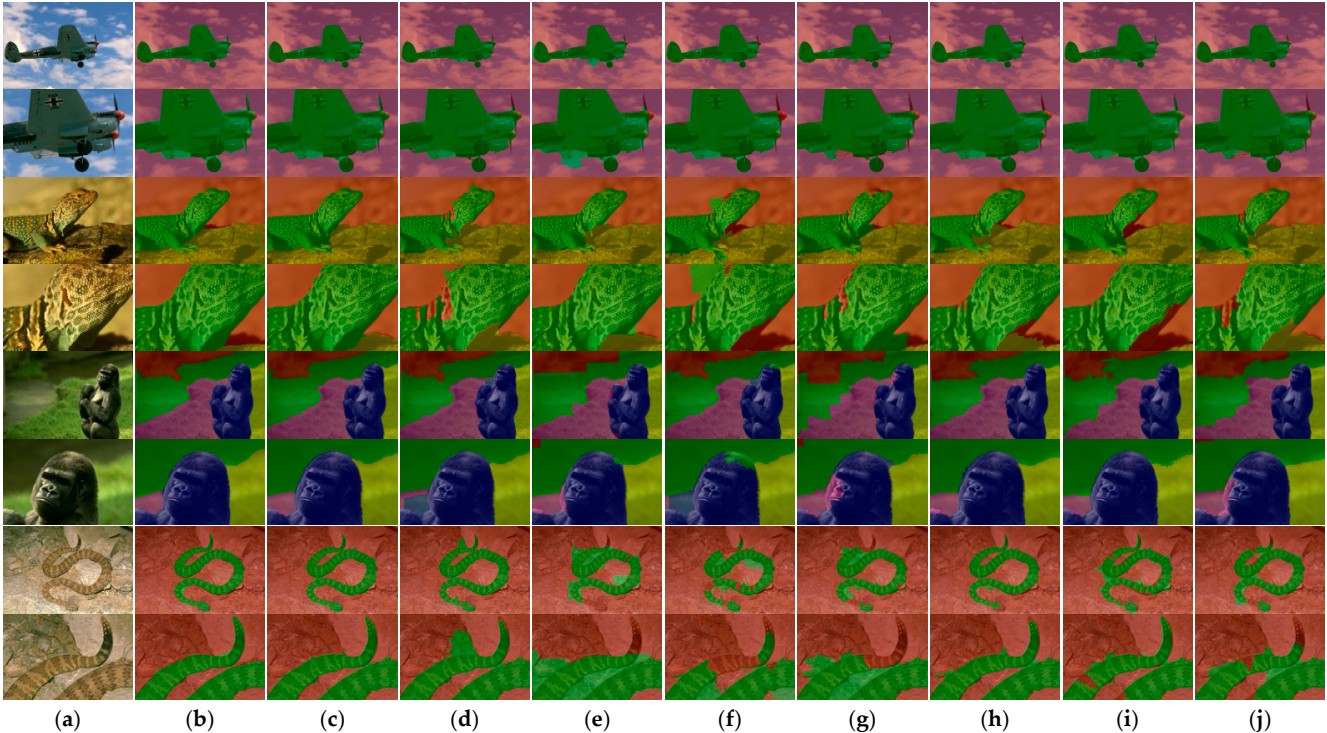

|   (a)   |   (b)   |   (c)   |   (d)   |   (e)   |   (f)   |   (g)   |   (h)   |   (i)   |   (j)   |

**Figure 11.** Achievable segmentation of superpixels on BSDS500. From left to right, (**a**) input image; (**b**) ground-truth segmentation; (**c**) CONIC; (**d**) SCALP; (**e**) TPS; (**f**) SLIC; (**g**) WS; (**h**) SEEDS; (**i**) ERS; (**j**) LRW. Alternating r show the local details of each image to facilitate close visual inspection.

It is worth noting in Figure 11 that the performance of CONIC is very similar to the hand-labeled ground truth. Moreover, the performance is similar to the metric evaluation of ASA in Figure 8c that gains insight into the quantitative results. Taking time efficiency into consideration, the proposed CONIC is more suitable for superpixel-based image segmentation among the state-of-the-art methods.

## 6. Conclusions

In this paper, a novel superpixel segmentation method termed Contour Optimized Non-Iterative Clustering (CONIC) is presented. Adopting the contour intensity as the prior information provides a balanced trade-off between segmentation accuracy and visual uniformity. The major improvement can be generalized into two aspects, including a seed redistribution strategy to promote the initialization step, and a subtle similarity measurement for the clustering step. Both optimizations generate a synergetic effect and perform significantly better than conventional SNIC and SCALP. Experimental results demonstrate that CONIC runs in a limited computational time with state-of-the-art performance on the public dataset.

Future work will focus on advanced computer vision applications based on the proposed algorithm. For example, since the CONIC superpixels could be more accurate

and regular in representing some artificial objects and homogeneous regions, it is more suitable to classify specific features from natural scenes.

**Author Contributions:** Conceptualization, investigation and writing-original draft preparation, C.L.; methodology and visualization, N.L.; software and funding acquisition X.H.; validation and resources, Z.C.; formal analysis and data curation, S.H.; writing-review and editing, J.G.; supervision, W.H.; project administration and funding acquisition, B.G. All authors have read and agreed to the published version of the manuscript.

**Funding:** This research is supported financially by National Natural Science Foundation of China (Grant No. 61972398 and 51805398) and Fundamental Research Funds for the Central Universities (Grant No. JB211303).

**Institutional Review Board Statement:** Not applicable.

**Informed Consent Statement:** Not applicable.

**Data Availability Statement:** The BSDS500 dataset and the reference codes in this work are available at: https://github.com/davidstutz/superpixel-benchmark (accessed on 29 January 2021).

**Acknowledgments:** The authors would like to thank the editor and anonymous reviewers for their valuable comments on this paper.

**Conflicts of Interest:** The authors declare no conflict of interest.

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
