# Peer review of "CONIC: Contour Optimized Non-Iterative Clustering Superpixel Segmentation"

_remotesensing, doi:10.3390/rs13061061_

Round 1

Reviewer 1 Report

This is referring to the study entitled "CONIC: Contour Optimized Non‐Iterative Clustering Super‐2pixel Segmentation.”

The authors developed algorithms for a more precise segmentation.

Minor comments:

  1. 34: How do the authors define the word “outstanding”?
  2. 41: A paper should be cited here.
  3. 43: How do the authors define “good superpixels”?
  4. 285: Where is the legend for figure 1e?

Major comments:

- The word “superpixel” should be explained easily understandable in the beginning of the introduction

- Of what can be seen in the figures, the image processing would surely create artifacts in terms of object intensity and thus probably in morphological analysis of identified objects.

- The authors may discuss the advantages of their method compared to normal thresholding or identification of an object after having created a binary image.

- The images depicted here are changed. Is it possible to analyze the original image with objects identified in the processed image?

- Many researches may want to analyze cell culture or even tissue. Some examples (i.e. figures) here would certainly be helpful.

Reviewer 2 Report

The paper presents a novel method that is correct and well described. The experiments are sustaining the conclusions of the theoretical approach.  There are no evident errors. 

As the paper was submitted to Remote Sensing it was expected that at least one experiment should have process some satellite data.  

Round 2

Reviewer 1 Report

This is referring to the study remotesensing-1110225. If I am not mistaken, I did not see comments of the authors. More important, my main concern about the study was completely ignored: Segmentation is a very important, if not the most important subject in image analysis. However,  the most readers may be interested in segmenting cells rather than living animals. The authors did not provide any example of cellular analysis. A comparison of cellular segmentation using established segmentation algorithms and the authors segmentation algorithm would make the study substantially more interesting and important.